# Pediatric Diffuse Midline Glioma H3K27-Altered: From Developmental Origins to Therapeutic Challenges

**DOI:** 10.3390/cancers16101814

**Published:** 2024-05-10

**Authors:** Manuela Mandorino, Ahana Maitra, Domenico Armenise, Olga Maria Baldelli, Morena Miciaccia, Savina Ferorelli, Maria Grazia Perrone, Antonio Scilimati

**Affiliations:** Research Laboratory for Woman and Child Health, Department of Pharmacy–Pharmaceutical Sciences, University of Bari “Aldo Moro”, Via E. Orabona 4, 70125 Bari, Italy; m.mandorino3@studenti.uniba.it (M.M.); ahana.maitra@uniba.it (A.M.); domenico.armenise1@uniba.it (D.A.); olga.baldelli@uniba.it (O.M.B.); morena.miciaccia@uniba.it (M.M.); savina.ferorelli@uniba.it (S.F.); mariagrazia.perrone@uniba.it (M.G.P.)

**Keywords:** diffuse intrinsic pontine glioma (DIPG), H3K27-altered, neonatal, brain development

## Abstract

**Simple Summary:**

Diffuse midline glioma (DMG), especially diffuse intrinsic pontine glioma (DIPG), is a deadly pediatric brain tumor that is difficult to diagnose and lacks any real treatment. These tumors often affect deep midline brain structures in young children, suggesting a connection to early brain development and differentiation. The H3K27M mutation triggers DIPG, impacting gene expression and brain development. Despite targeted drug interventions for gene mutations, the grim prognosis of the disease remains unaltered. DIPG patients typically succumb to the illness within 12 to 18 months post-diagnosis. Our review found over 85% of DIPG tumors have the K27M mutation in histone genes, driving abnormal growth. This mutation affects crucial brain processes, including the epithelial–mesenchymal transition pathway, potentially explaining differences between gliomas with and without K27M. The timing of these mutations is not known. One idea is that these mutations might have started during the early development of the brain before birth.

**Abstract:**

Diffuse intrinsic pontine glioma (DIPG), now referred to as diffuse midline glioma (DMG), is a highly aggressive pediatric cancer primarily affecting children aged 4 to 9 years old. Despite the research and clinical trials conducted to identify a possible treatment for DIPG, no effective drug is currently available. These tumors often affect deep midline brain structures in young children, suggesting a connection to early brain development’s epigenetic regulation targets, possibly affecting neural progenitor functions and differentiation. The H3K27M mutation is a known DIPG trigger, but the exact mechanisms beyond epigenetic regulation remain unclear. After thoroughly examining the available literature, we found that over 85% of DIPG tumors contain a somatic missense mutation, K27M, in genes encoding histone H3.3 and H3.1, leading to abnormal gene expression that drives tumor growth and spread. This mutation impacts crucial brain development processes, including the epithelial–mesenchymal transition (EMT) pathway, and may explain differences between H3K27M and non-K27M pediatric gliomas. Effects on stem cells show increased proliferation and disrupted differentiation. The genomic organization of H3 gene family members in the developing brain has revealed variations in their expression patterns. All these observations suggest a need for global efforts to understand developmental origins and potential treatments.

## 1. Introduction

Diffuse intrinsic pontine glioma (DIPG), now more commonly referred to as diffuse midline glioma (DMG), is an aggressive and universally fatal pediatric cancer originating from glial cells surrounding and safeguarding neurons in the brain [1,2]. This devastating disease predominantly affects children in age groups 0–4 and 5–9; girls have a worse prognosis than boys with DMGs [3,4] (Figure 1). They are typically located in the pons, which is a brainstem region crucial for breathing, heartbeat, and consciousness. With approximately 150–400 new cases diagnosed annually in the United States [4] and a similar number in Europe (Figure 1), common symptoms include coordination problems, limb weakness, speech difficulties, and vision impairments. DIPG also presents a diagnostic challenge as it cannot be surgically removed due to its critical location [5].

The standard treatment for DIPG is focal radiation, which may provide temporary relief from symptoms but does not impact the overall survival of DIPG patients [6,7,8,9]. Unfortunately, despite treatment efforts, the average survival for DIPG patients is twelve months with a five-year survival rate below 5% [8]. Over more than 40 years of clinical trials, there has been no improvement in the overall survival rate for DIPG [10]. Fractioned radiotherapy (59 Gray) alleviates disease symptoms in DIPG cases. Approximately 85% of DIPG patients exhibit H3K27-altered mutations along with concurrent genetic mutations. Despite targeted drug interventions for gene mutations, the grim prognosis of the disease remains unaltered. DIPG patients typically succumb to the illness within 12 to 18 months post-diagnosis. The timing of these gene alterations remains unknown, and a plausible hypothesis suggests that these mutations may have originated during embryonic brain development.

Understanding DIPG biology was historically limited due to the scarcity of available tumor tissue for molecular and genetic analysis. Nowadays, tumor biopsy execution has become an almost widely used protocol due to the increased availability of technology. As a consequence, recent genomic investigations of pediatric DIPG and thalamic glioma tissue samples have shown that approximately 85% of DIPG tumors display a distinctive mutation known as lysine 27 to methionine (K27M) in genes encoding histone H3.3 and H3.1, leading to abnormal transcription [11,12,13,14,15,16,17]. This mutation induces a structural alteration in the histone protein, disrupting its capacity to interact with DNA and other proteins. As a result, changes in gene expression facilitate the proliferation and spread of cancer cells [18].

Recognizing the significant impact of this mutation on tumor biology and clinical outcomes, the tumor was reclassified as “H3K27-altered” in the 2021 WHO Classification of Tumors of the Central Nervous System, departing from the previous terminology “H3 K27M-mutant, WHO grade IV” [19,20]. It is unknown at what period of the child’s life the alteration of H3K27 occurs. As a hypothesis, it could happen during neural development. Hence, besides recovering known data, the main purpose of this review is to stimulate further worldwide efforts.

## 2. Material and Methods

A comprehensive search of the literature from 1993 to 2023 was conducted for this narrative review using PubMed database (https://pubmed.ncbi.nlm.nih.gov/) (accessed on 24 April 2024). Controlled search terms were used to gather relevant articles on various aspects related to DIPG and brain development. The search included specific queries such as “DIPG/H3K27M and brain development” (16 articles), “DIPG/brain development/RNA sequencing” (10 articles), “DIPG mutation/neonatal” (9 articles), “DMG mutation/neonatal” (4 articles), “RNA sequencing/DIPG/children” (7 articles), “DIPG/OPC” (2 articles), and “DMG/OPC” (3 articles). Studies that appeared in multiple searches were considered only once. Then, the publications were evaluated for relevance. After careful investigation, 42 studies were deemed appropriate for this narrative review, primarily focusing on the tumors associated with the pons, as DIPG is a dismal cancer type. Key findings and insights on associated epigenetic regulations, concurrent mutations, early brain development, and future perspectives for therapeutic strategies were extracted and synthesized to construct a coherent narrative.

## 3. H3K27-Altered DIPG and Brain Development

More than 85% of DIPGs contain a somatic missense mutation in the genes responsible for encoding Histone 3 (isoforms H3.3 and H3.1), which changes lysine 27 with methionine. Tumors with H3.1 mutations differ from those with H3.3 mutations because they occur at a younger age and have distinct clinicopathologic and radiologic features, which include a slightly longer survival time [21].

However, although the H3K27M mutation predominantly arises in the H3F3A locus encoding histone variant H3.3, it is found in approximately 20% of DIPG cases in H3.1 or H3.2 variants, typically within the HIST1H3B gene [16,22,23,24,25]. Very often, each type of H3K27M DIPG acquires distinct secondary mutations. Particularly when the H3.3K27M mutation occurs in DIPG tumors, it often coincides with mutations in the TP53 pathway. TP53 is a tumor-suppressor gene involved in regulating cell growth and division.

Therefore, mutations in the TP53 pathway alongside H3.3K27M mutations may indicate a combined effect contributing to the development or progression of DIPG. On the other hand, when the H3.1K27M mutation is present in DIPG tumors, it frequently occurs alongside mutations in genes associated with the ACVR1 and phosphatidylinositol 3-kinase (PI3K) pathways. ACVR1 is a gene encoding a receptor involved in cell signaling, while the PI3K pathway regulates various cellular processes, including cell growth and survival [16,22,23,24,25] (Figure 2).

The human brain experiences rapid growth during pregnancy from 4 to 24 weeks after conception. Neuronal precursor cells (NPCs), which later become neurons, divide at an astonishing rate of over 105 divisions per minute, emphasizing the complexity of brain development during this period [26]. After birth, the subventricular zone in the brain becomes active, generating new neurons through neurogenesis. These neurons migrate to the prefrontal cortex, which governs complex cognitive functions like decision making and social behavior [27]. Additionally, evidence suggests that somatic mutations occur in the developing human brain alongside the continuous process of neurogenesis [28,29].

### 3.1. Histone H3 Dynamics

A recent in silico study [30] examined the genomic organization and expression of the human H3 gene family during brain development using publicly available RNA-seq-based datasets. The research findings indicated significant transcriptional activity, which is the process of gene expression through the creation of RNA molecules, from at least 17 genes encoding histone H3 proteins in the developing brain. These genes produce various forms of histone H3 proteins, including the traditional (canonical) type known as H3.1 and a type called replication-independent H3.3. In total, six variations of histone H3 proteins were identified. One notable finding was that the genes responsible for producing H3.3 proteins, such as H3F3A, showed a gradual decrease in activity as the brain continued to develop. This suggested that the production of H3.3 histone proteins decreases over time as the brain matures.

Conversely, genes encoding H3.1 proteins, like HIST1H3B, demonstrated a different pattern. They observed significant downregulation, or reduced activity, during the early stages of prenatal brain development. After this initial decrease, these genes remained mostly inactive or silent. These results indicated distinct regulatory patterns governing the production of different histone H3 proteins during brain development. The gradual decrease in H3.3 production suggests a potential role for these proteins in early developmental processes, while the downregulation of H3.1 genes implies a shift in the types of histone proteins utilized as brain development progresses.

Additionally, it was observed that some H3 genes contained a mutable codon, K27-AAG, while others had the alternative codon, AAA, at this position. HIST1H3B, a member of the H3.1 class, constituted the largest proportion of H3.1 transcripts among H3.1 isoforms in the early developing human brain. Variations in clinical characteristics and DNA methylation patterns have been documented in DIPGs with H3.1K27M and H3.3K27M mutations [21,31,32]. Nonetheless, the impact of H3K27M seems to rely on both the chromatin surroundings and the oncohistone’s presence in chromatin. 

### 3.2. Impacts of H3.3K27M and H3.1K27M Oncohistones in DIPG Tumorigenesis

Nagaraja et al. [33] investigated the molecular differences between H3.3K27M and H3.1K27M oncohistones regarding chromatin dysregulation and tumorigenesis. They performed H3K27ac chromatin immunoprecipitation sequencing (ChIP-seq) analysis on twenty-five post-mortem DIPG tumor samples, which included sixteen H3.3K27M DIPGs, nine H3.1K27M DIPGs and five rare normal pediatric pontine tissue samples from non-malignant pediatric pons specimens. They found that H3.3K27M DIPG exhibited an enrichment of transcription factors associated with early neural development. This notably included the Regulatory Factor binding to the X-box (RFX) family, which plays a pivotal role in forming midline brain structures [34].

In contrast, H3.1K27M DIPG displayed an enrichment of motifs related to Nuclear Factor Erythroid 2 (NFE2) signaling, particularly involving NFE2L3, which is recognized for its involvement in mediating cancer cell resistance to therapy [35]. Furthermore, these findings highlight differences in the localization patterns of two variants of histone proteins, H3.3K27M and H3.1K27M, within the genome compared to their normal forms and their potential implications for chromatin structure and gene regulation in DIPG. Specifically, H3.3K27M is typically found in regions of active chromatin, which are areas of the genome where gene expression is actively regulated. In contrast, H3.1K27M is distributed more widely throughout the genome, suggesting it may have a broader impact on gene regulation [36]. 

This difference in patterns of localization may have important implications. For instance, it could influence how these variants affect histone modifications (chemical changes to histone proteins that can influence gene expression), the accessibility of chromatin (how easily DNA can be accessed and transcribed), and the binding of transcription factors (proteins that regulate gene expression) in different subgroups of DIPG. Despite the H3K27M mutation representing only a small fraction (3–17%) of the entire histone H3 protein, it significantly impacts the epigenetic landscape of DIPG [14]. Epigenetic dysregulation stands out as a significant hallmark of childhood cancers, where malignancy is linked to the extensive disruption of gene expression. One crucial epigenetic change caused by the H3K27M mutation is the reduction in H3K27me3 levels, which is a specific histone modification that plays a vital role in gene regulation through post-translational modification. The depletion of H3K27me3 is attributed to the disruption of normal polycomb repressive complex 2 (PRC2) function by H3K27M, although various alternative mechanisms have been suggested [37,38,39].

H3K27me3 is a critical player in epigenetic regulation, which controls gene activation or repression without altering the underlying DNA sequence [14,40,41]. In DIPG, the extensive perturbation of epigenetic control promotes the development of cancer, and in certain instances, this is accompanied by secondary mutations in well-established oncogenic pathways [11,16,42]. 

### 3.3. EMT and H3K27me3 Dysregulation in Gliomagenesis

Numerous crucial processes in brain development are controlled by H3K27me3 deposition, and their dysregulation potentially contributes to gliomagenesis. One such biological process is the epithelial–mesenchymal transition (EMT) pathway, which is vital for several key developmental events in early embryonic development [43,44,45] (Figure 2).

Firstly, it plays a crucial role in gastrulation, which is a pivotal stage early in embryonic development where the embryo transforms from a simple spherical structure into a more complex, multi-layered structure with distinct tissue layers. EMT is involved in the movement and rearrangement of cells during this process, allowing for the formation of these tissue layers. 

Secondly, the EMT pathway is essential for neural crest cell migration. Neural crest cells are a group of cells that migrate from the developing neural tube (which gives rise to the brain and spinal cord) to various locations throughout the embryo, where they differentiate into a diverse array of cell types, including neurons, glial cells, and cells of the peripheral nervous system. The EMT pathway facilitates the migration of these cells from their site of origin to their final destinations.

Lastly, EMT also plays a critical role in neural tube formation. The neural tube is the precursor to the central nervous system, including the brain and spinal cord. During embryonic development, the neural tube forms through a complex series of morphogenetic processes, one of which involves EMT. This pathway helps shape and structure the neural tube, ensuring proper nervous system development. 

Sanders et al. [46] conducted a comprehensive analysis using RNA sequencing datasets to compare gene expression patterns between pediatric gliomas with H3K27M and those without this mutation (referred to as non-K27M tumors). Their findings revealed intriguing insights into the role of the EMT pathway in these tumors. They observed that genes associated with EMT were significantly enriched among the genes that displayed differential expression between H3K27M and non-K27M tumors. Specifically, the tumors with the H3K27M mutation showed an increased expression of genes associated with the early stages of EMT (pre-EMT genes) and a decreased expression of genes linked to the later stages of EMT (post-EMT genes) compared to non-K27 M tumors. Furthermore, the analysis of cerebral organoid data—a model system that mimics the structure and function of the human brain—supported the idea that H3K27M tumors share similarities with normal brain cells in the pre-EMT stage. This suggests that the molecular characteristics of H3K27M tumors may resemble those of normal brain cells undergoing the initial stages of EMT. 

Moreover, the researchers examined a dataset generated through single-cell RNA sequencing, which is a powerful technique that analyzes gene expression in individual cells. This analysis revealed the presence of different stages of EMT in both H3K27M and non-K27M gliomas. Interestingly, they found that tumors with a mutation in the H3.1 histone variant (H3.1K27M) appeared to resemble a later stage of EMT compared to tumors with a mutation in the H3.3 variant (H3.3K27M). In DIPG tumors, there are alterations in the levels of H3K27me3, which may disrupt its normal suppressive function. This disruption could lead to the increased expression of genes associated with cancer development. 

Consequently, these changes in gene expression could contribute to the initiation and progression of DIPG, potentially fueling the aggressive nature of this type of brain cancer [47]. The global loss of H3K27me3 in H3K27M-altered gliomas is thought to deregulate gene expression, potentially leading to tumorigenesis. Oncogenesis may occur only when the mutation occurs within a cell in a susceptible transcriptional state, such as early neural or glial precursor cells [42,48,49,50,51].

The occurrence and distribution of DIPG tumors are not random but rather follow specific spatial and temporal patterns [1]. This observation suggests that the formation of these tumors may be influenced by factors related to the surrounding environment and the timing of cellular events during brain development. Overall, the development of DIPG tumors may be intricately linked to disruptions in normal neurodevelopmental processes.

### 3.4. H3K27M and Oligodendroglial Precursor Cells (OPCs)

Tumor cells could potentially originate from a susceptible type of cell (referred to as the “cell of origin”) within the brainstem, and these cells may be influenced by cues from their microenvironment that support or promote tumor development.

Furthermore, histological findings have indicated that DIPG tumors often appear in regions of the brainstem where early oligodendroglial precursor cells are abundant. These cells are part of the oligodendroglial lineage, which is responsible for producing myelin—a fatty substance that insulates nerve fibers and facilitates efficient neural communication. Oligodendroglial precursor cells (OPCs) are crucial for myelin development, particularly during childhood and adolescence.

The proliferation and viability of OPCs rely on the activation of platelet-derived growth factor A (PDGF-A) and its corresponding receptor, PDGFRA. However, while PDGFRA signaling is essential for maintaining OPCs, it acts as a deterrent to their differentiation process. Notably, PDGFRA signaling diminishes as OPCs progress toward maturation into myelinating oligodendrocytes (OLs) [48]. In contrast, PDGFRA is frequently subject to genetic amplification or mutation in numerous glioma subtypes, including DMG [22]. Nonetheless, the specific cellular and molecular alterations triggered within OPCs in response to dysregulated PDGFRA expression remain poorly understood.

Cardona et al. [49] generated mice harboring a conditional knock-in (KI) of wild-type human PDGFRA (*h*PDGFRA), which is selectively upregulated in prenatal Olig2- or GFAP-expressing progenitors (pivotal brain cell precursors). Their investigation revealed that the prenatal overexpression of *h*PDGFRA in glial progenitors results in impaired oligodendroglial development and subsequent hypomyelination in the central nervous system.

Emerging evidence posits neonatal brainstem OPCs as likely origins of DIPG [33,50]. This assertion is supported by earlier studies revealing a spatial and temporal correlation between the proliferation of Olig2-expressing precursor cells in human and murine brainstems and the onset of brainstem gliomas in pediatric patients [51]. These findings underscore the potential involvement of OPCs in the pathogenesis of this cerebral tumor.

Although the suspected origination of DIPG from OPCs in the brainstem has lacked experimental validation through a representative mouse model, Tomita et al. [52] addressed this gap by employing the RCAS/Tv-a avian retroviral system to instigate the formation of DMGs within two distinct populations of brain progenitor cells, those expressing Olig2 and those expressing Nestin, within neonatal mouse brainstems. Gliomas were successfully induced in both cellular models by introducing specific genetic manipulations, including PDGF-A or PDGF-B overexpression and p53 deletion.

Notably, upon introducing the H3.3K27M mutation, divergent outcomes were observed based on the cellular context. In Nestin-expressing cells, this mutation markedly expedited tumor progression and enhanced cellular proliferation compared to wild-type H3.3, whereas its impact was less pronounced in Olig2-expressing cells. Furthermore, the frequency of H3.3K27M-expressing cells was diminished in tumors originating from Olig2-expressing cells relative to Nestin-expressing cells, suggesting the varied necessity for this mutation contingent upon the cell type of origin.

Subsequent RNA-sequencing analysis unveiled distinct transcriptional profiles in tumors derived from differing cell types and genetic modifications. Through gene set enrichment analysis (GSEA), cell-of-origin-specific ramifications of H3.3K27M were delineated, promoting EMT and angiogenesis in Olig2-marked tumors while attenuating these processes in Nestin-marked tumors. These findings underscore the notion that the oncogenic influence of the H3.3K27M mutation exhibits variability contingent upon the specific progenitor cell type with Nestin-expressing cells displaying heightened susceptibility to its effects relative to Olig2-expressing cells.

There is a protein called Tenascin-C (TNC) that plays different roles in the context of brain development, including guiding neuron migration, maintaining the stem cell niche, and potentially influencing the behavior of these precursor cells, possibly contributing to the initiation or progression of DIPG [53]. TNC is described as an extracellular matrix (ECM) glycoprotein. The ECM is a complex network of proteins and other molecules that provide structural support and regulate various cellular functions. Glycoproteins are proteins that have sugar molecules attached to them. 

TNC is pivotal in mediating interactions between cells (cell–cell) and between cells and their surrounding environment (cell–matrix), which is essential for processes like cell migration, adhesion, and signaling. Additionally, TNC is involved in guiding migrating neurons during normal brain development, indicating its regulatory function in directing neurons to appropriate locations to form proper neural circuits and brain architecture. Moreover, TNC contributes to maintaining a specialized microenvironment known as a stem cell niche in the developing brain, where it supports the survival and proliferation of stem cells by modulating the activity of signaling pathways such as Platelet-Derived Growth Factor (PDGF) and Notch, which regulate cell growth and differentiation.

Qi et al. [54] have analyzed TNC expression patterns, investigating their associations with clinicopathological features and exploring the biological effects of TNC in pediatric gliomas, including DIPG. Firstly, they found significantly increased TNC expression in DIPG tumor tissue compared to normal brain tissue. They confirmed this result by analyzing a large cohort of pediatric glioma specimens and mouse xenograft tumors. Moreover, they observed that high TNC expression extent was associated with higher tumor grade and poorer clinical outcomes, including overall survival and tumor recurrence, suggesting a potential role for TNC in tumor progression. Further investigations using cell lines revealed that greater TNC expression was linked to the presence of the H3K27M mutation. Interestingly, TNC also exhibited differential effects on cell proliferation depending on the presence of the H3K27M mutation. 

DIPG develops during active midline myelin structure formation, but it is unclear when and in which cell type the H3K27M mutation occurs. Recent genetic mouse model studies suggest that genetic alterations commonly associated with midline gliomas, including the H3K27M mutation, can induce tumor formation in the brainstem during the postnatal period if introduced into the brain during prenatal brain development. This highlights the critical role of prenatal events in the development of these types of brain tumors [55]. Additionally, when the H3K27M mutation is introduced into neural stem cells (NSCs) cultured in a lab in combination with other genetic changes, such as TP53 gene knockout and/or PDGFRA gene amplification and after their transplantation into mice brains, surprisingly, the resulting tumors did not resemble the typical appearance of diffuse midline gliomas when observed under a microscope. The tumors lacked the expected characteristics seen in patients with diffuse midline gliomas [56].

This finding suggests that while the introduced mutation and other genetic alterations might have played a role in tumor formation, they alone were not enough to create tumors resembling diffuse midline gliomas. This implies the involvement of additional factors or genetic changes required to replicate these brain tumors’ features fully. There is evidence supporting the idea that brain tumors may develop according to a model where mutations initially occur in NSCs, leading to the formation of cancer arising from more specialized neural precursor cells. This model is supported by findings from studies conducted on adult glioblastoma multiforme (GBM) tumors. They have shown that mutations associated with this cancer when introduced into NSCs in genetic mouse models lead to tumor formation only when the stem cell differentiates into a specific precursor cell called OPC [57]. Further research suggests that GBMs may arise from both NSCs and OPCs. The specific molecular subtype of GBM that develops may be determined by the type of precursor cell from which the tumor originates [58]. The NSC model of glioblastoma tumorigenesis is further supported by single-cell RNA sequencing analysis findings, which revealed that the cells primarily responsible for driving the disease in patient tumors express gene signatures similar to OPCs. This finding strengthens the evidence supporting the involvement of NSCs in tumor initiation and progression [54]. Therefore, in the context of brain development or in various neurological processes, including tumor formation, possible cell types that could be involved are neuroepithelial cells (NSCs), radial glia (neural progenitor cells), and OPCs [44,53,54].

In their study, Silveira et al. [59] investigated the implications of the histone H3K27M mutation in DIPG tumors. Utilizing DIPG xenograft models—tumors developed in mice from human DIPG cells—they conducted experiments to suppress the expression of this mutation. This intervention resulted in the restoration of H3K27me3 levels. Comparative analysis between DIPG xenografts with and without knockdown of the H3K27M mutation unveiled mutation-specific effects on both the transcriptome (the complete set of RNA molecules in a cell) and epigenome (the overall pattern of chemical modifications to DNA and histone proteins). Specifically, the mutation was found to directly influence the expression of certain genes by releasing poised promoters, thereby instigating alterations in gene expression patterns implicated in proliferation and differentiation processes, ultimately contributing to the tumor’s phenotype and growth.

In a subsequent study, Haag et al. [60] investigated the effects of the H3.3K27M mutation associated with DIPGs using human induced pluripotent stem cells (iPSCs) engineered to carry this mutation. Employing an inducible form of the H3.3K27M mutation, they strategically introduced it into the iPSC genome, allowing precise control over its activation. The researchers then scrutinized the impact of this mutation on various neural cell types derived from these iPSCs. Notably, the mutation was found to enhance the expression of developmental genes, particularly those situated at bivalent promoters—regions of DNA capable of activating or repressing genes crucial for maintaining cellular identity and function. Interestingly, the effects of the mutation varied across different neural cell types: iPSCs bearing the mutation exhibited compromised viability, while NSCs displayed augmented proliferation upon induction of the mutation with OPCs also exhibiting a moderate increase in proliferation. Remarkably, when the mutation was coupled with the inactivation of TP53 in NSCs, tumor formation occurred upon transplantation into mice, effectively mirroring human DIPGs in an orthotopic xenograft model. This underscores the pivotal role of the H3.3K27M mutation and TP53 in NSCs initiating tumorigenesis. Specifically, in NSCs, the H3.3K27M mutation sustained the expression of genes associated with stemness and proliferation while prematurely activating OPC-related genetic programs. These molecular alterations likely underlie the initiation of tumor formation by fostering aberrant cell growth and impeding normal cellular differentiation processes.

To evaluate the impact of this mutation on stem cell state and differentiation potential without the influence of other mutations, Kfoury-Beaumont et al. [61] studied how the H3K27M mutation affects cellular proliferation and differentiation in human embryonic stem cell models, creating a human embryonic stem cell (*h*ESC) line by introducing a single base mutation (A>T) into the H3F3A gene. This mutation led to an amino acid substitution from K to M at position 27 in the protein, resulting in H3K27M. They found that H3K27M enhances stem cell proliferation, disrupting differentiation and causing abnormal gene expression during cell specification. This mutation leads to a partially differentiated state with increased clonogenicity, potentially creating conditions favorable for acquiring additional mutations that cooperate in gliomagenesis.

## 4. Conclusions

In conclusion, mutations in genes encoding histone H3 variants, notably the H3K27M mutation, are important in driving DIPG development. Despite their prevalence across most DIPG cases, the disease exhibits considerable clinical and molecular diversity, which is marked by secondary mutations. Concurrent mutations in key signaling pathways like TP53, ACVR1, and PI3K emphasize the complex molecular landscape of DIPG tumorigenesis. These mutations disrupt histone protein distribution, impacting gene regulation and chromatin structure. This disruption, coupled with the altered function of epigenetic regulators like PRC2, leads to aberrant gene expression patterns, promoting cancer development. Understanding these molecular dynamics is essential for creating effective DIPG therapies [62].

Targeting the epigenetic dysregulation caused by H3K27M mutations presents a promising therapeutic strategy. HDAC inhibitors and other epigenetic-modifying agents hold the potential for restoring normal histone modifications and gene expression patterns disrupted by H3K27M mutations. Additionally, exploiting synergistic interactions between H3K27M mutations and radiotherapy may enhance treatment efficacy. Further research into the molecular mechanisms underlying DIPG tumorigenesis, including the role of Tenascin-C in tumor progression and the influence of H3K27M mutations on oligodendroglial precursor cells, is necessary to identify novel therapeutic targets and improve patient outcomes. Ultimately, it is really important to understand the genetic and epigenetic changes that cause DIPG to create treatments that are personalized and specifically target this fatal pediatric brain cancer.

## Figures and Tables

**Figure 1 cancers-16-01814-f001:**
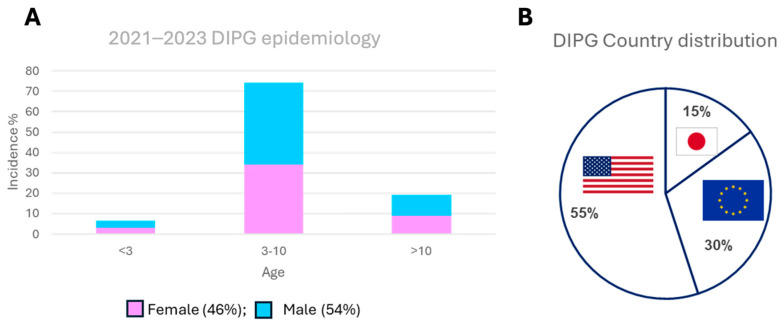
(**A**) DIPG age and gender distribution of 1252 cases in 2021–2023 adapted from DIPG Registry https://www.dipgregistry.org (blue represents male and pink represents female); (**B**) DIPG country distribution adapted from https://www.imarcgroup.com/diffuse-intrinsic-pontine-glioma-market (accessed on 3 May 2024).

**Figure 2 cancers-16-01814-f002:**
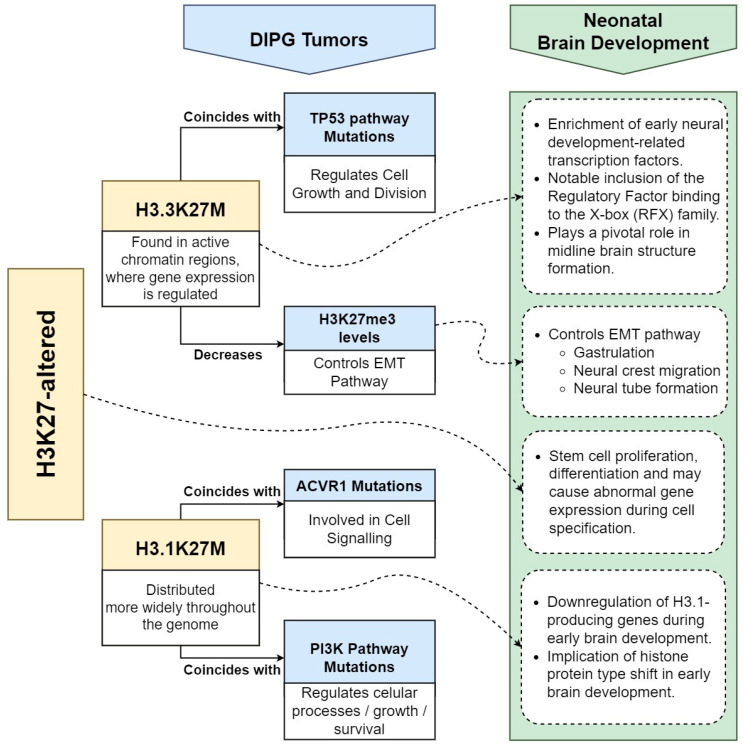
High-level schematics presenting the different regulatory patterns and associated metabolic pathways of H3K27-altered and its two variants (H3.3K27M and H3.1K27M) concerning DIPG tumors and neonatal brain development.

## Data Availability

Data sharing does not apply to this article, as no datasets were generated or analyzed during the current study.

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
