# Peer review of "Pediatric Diffuse Midline Glioma H3K27-Altered: From Developmental Origins to Therapeutic Challenges"

_cancers, 2024, doi:10.3390/cancers16101814_

Round 1

Reviewer 1 Report

Comments and Suggestions for Authors

The authors present a review of the literature on neonatal genetic mutations that are implicated in diffuse intrinsic pontine glioma/diffuse midline glioma initiation and progression. Their focus is on the H3K27M mutation and its role in disease onset. It is an interesting read and the topic is of interest and importance. Overall it is well written, although some revision is required before acceptance for publication.

In the introduction, the authors state that DIPG is a “very often fatal” paediatric cancer. To my knowledge, it is “universally fatal”. They continue to state that predominantly children aged 4 – 9 are affected, irrespective of gender. A recent study by Hoogendijk et al. has shown that in age groups 0 – 4 and  5 – 9, girls have a worse prognosis than boys with DMGs. Maybe this can be mentioned.  

At the beginning of the introduction, diffuse intrinsic pontine glioma has been abbreviated to (DIPG). The authors use this term and then abbreviate again in the first paragraph following figure 1. The authors should revise how they abbreviate. On page 6, they abbreviate oligodendroglial precursor cells to OPCs after the second time they mention this cell type. They also abbreviate neural stem cells to NSCs twice on page 7. Maybe a list of abbreviations in a table form could be of use, given the large number of abbreviations used.

On page 4, change (NFE2] to (NFE2)

On page 5, change [EMT) to (EMT), and edit the sentence for grammar.

Consistency. On page 5/6, either refer to mutations as H3.1K27M or H3.1-K27M, but nor both. Also applies to H3.3.

On page 5, in the paragraph beginning “Sanders et al;, [50]”, remove the word called. It reads as if this mutation is being introduced to the reader at this point of the review. Rewrite to state that a comparison was made between paediatric gliomas harbouring the H3K27M mutation and those without.

On page 6, the paragraph “the occurrence and distribution of DIPG…” lack references.

On page 6, the role of TNC is mentioned in both the paragraphs beginning “interestingly” and “TNC mediates”. These two paragraphs can be combined to improve clarity and avoid stating the same thing in different ways.

On page 7, the sentence beginning ”Silveira et al. [60], researchers” needs to be rewritten for clarity.

On page 7, last paragraph, the authors describe research by Haag et al, but the way it is written needs to be adjusted. They state “this allows us to control when..” but then talk about how “they studied the effects..”. Reword for clarity.

In the materials and methods, the authors could elaborate on the search terms they chose for the literature search. They state that DIPG is now more commonly referred to as DMG, yet do not use “DMG” in their search terms. For example, DIPG/OPC returned 2 articles, while DMG/OPC returns 3. One of these studies (by Cardona et al., 2021) is relevant for this study. Substituting the term DIPG with DMG could give additional articles. All search terms returned the stated articles (16, 9 and 2) except for the search term “DIPG/brain development/RNAseq” (3). Why not include DIPG/EMT, or other combinations to ensure all critical articles are identified? Also, what is the criteria for “irrelevant” articles that are then excluded? Please state how many articles were found in total, and how many were included/excluded.

The conclusion section is inadequate and needs to be expanded appropriately. I don’t see why the authors jump to pharmacodynamic/pharmacokinetic profiles of drugs, and drug delivery methods such as CED and focused ultrasound, and challenges such as traversing the blood brain barrier. They should elaborate on the mutations and pathways that they have discussed in the review, and suggest viable options to target these mutations, for example the use of HDAC inhibitors, exploiting mutations to synergise with IR therapy, etc. Then they can address other challenges such as delivery and BBB permeability.

Comments on the Quality of English Language

Overall, the English is sound, needing only minor revisions for grammatical correctness. Some sentences need to be restructured for clarity. 

Author Response

The answer has been provided to all comments received.

Reviewer 2 Report

Comments and Suggestions for Authors

Dear authors,

This article summarizes the current knowledge of the plausible molecular mechanism of DIPG and the plausible role of K27M mutation in H3.1 and H3.3 in the onset of this disease. Please consider following suggestive comments:

1. The title of this article is misleading. Unless you have solid evidences to support the statement, neonate mutation is not a correct statement for this article. It is still unknown when the mutation occurs prior to the disease onset.

2. Regarding section 3, since K27M belongs to somatic mutation and the mechanism described is the research result from glioma, the part for adult brain development is really irrelevant (Ref. 26-33). Please provide your rationale of this part.

3. Consider to have subtitles for each research aspect of DIPG and H3.1 or H3.3 associated pathways in section 3 for clarification.

Author Response

The answer has been provided to all comments.

Reviewer 3 Report

Comments and Suggestions for Authors

This review is not well organized, please make the following modifications.

1. Please add subtitles to the third section (3. H3K27-Altered DIPG and Brain Development),

for example, 3.1 H3K27M mutation---, 3.2 H3.1K27M---, 3.3 H3.2K27M---, 3.4 H3.3K27M---, 3.5 EMT and H3K27M, 3.6 H3K27M and Oligodendroglial precursor cells (OPCs), 3.7 H3K27M and stem cells.

2. In the conclusion, needs to clarify the medical significance of H3K27M, for example, whether it is used for precise diagnosis or for drugs based on H3K27M, and summarizes the conclusion in one paragraph.

Author Response

Tha answer has been provided to all comments received.

Round 2

Reviewer 3 Report

Comments and Suggestions for Authors

The author has made appropriate revisions to the paper.